# Sarcopenia in Neurological Patients: Standard Values for Temporal Muscle Thickness and Muscle Strength Evaluation

**DOI:** 10.3390/jcm9051272

**Published:** 2020-04-28

**Authors:** Ariane Steindl, Johannes Leitner, Matthias Schwarz, Karl-Heinz Nenning, Ulrika Asenbaum, Sophie Mayer, Ramona Woitek, Michael Weber, Veronika Schöpf, Anna S. Berghoff, Thomas Berger, Georg Widhalm, Daniela Prayer, Matthias Preusser, Julia Furtner

**Affiliations:** 1Division of Oncology, Department of Medicine I, Medical University of Vienna, 1090 Vienna, Austria; 2Department of Biomedical Imaging and Image-guided Therapy, Medical University of Vienna, 1090 Vienna, Austria; 3Department of Neurology, Medical University of Vienna, 1090 Vienna, Austria; 4Department of Neurosurgery, Medical University of Vienna, 1090 Vienna, Austria

**Keywords:** sarcopenia, cranial MRI, temporal muscle thickness, reference values, muscle strength

## Abstract

Temporal muscle thickness (TMT) was investigated as a novel surrogate marker on MRI examinations of the brain, to detect patients who may be at risk for sarcopenia. TMT was analyzed in a retrospective, normal collective cohort (*n* = 624), to establish standard reference values. These reference values were correlated with grip strength measurements and body mass index (BMI) in 422 healthy volunteers and validated in a prospective cohort (*n* = 130) of patients with various neurological disorders. Pearson correlation revealed a strong association between TMT and grip strength (retrospective cohort, ρ = 0.746; *p* < 0.001; prospective cohort, ρ = 0.649; *p* < 0.001). A low or no association was found between TMT and age (retrospective cohort, R^2^ correlation coefficient 0.20; *p* < 0.001; prospective cohort, ρ = −0.199; *p* = 0.023), or BMI (retrospective cohort, ρ = 0.116; *p* = 0.042; prospective cohort, ρ = 0.227; *p* = 0.009), respectively. Male patients with temporal wasting and unintended weight loss, respectively, showed significantly lower TMT values (*p* = 0.04 and *p* = 0.015, unpaired *t*-test). TMT showed a high correlation with muscle strength in healthy individuals and in patients with various neurological disorders. Therefore, TMT should be integrated into the diagnostic workup of neurological patients, to prevent, delay, or treat sarcopenia.

## 1. Introduction 

Sarcopenia is a generalized skeletal muscle disease with a progressive course, commonly found in older individuals or systemic disease [1,2]. Based on the updated recommendations in 2018 of the European Working Group on Sarcopenia in Older People (EWGOP), the definition of sarcopenia comprises three different diagnostic features: (1) muscle strength; (2) muscle quantity; and (3) physical performance [2]. Muscle strength, represented by hand-grip strength, is reported to be currently the most reliable assessment of muscle function, due to its simplicity and its low cost [2]. Although the awareness of the importance of the diagnosis and care for patients suffering from sarcopenia is high due to the associated increased individual and socioeconomic burden, the evaluation of the loss of muscle strength and muscle quantity or quality is still not integrated into the routine clinical workflow [3,4,5,6,7]. This may be due to the fact that the assessment of skeletal muscle mass and function is based on additional examinations that result in a higher radiation dose for the patient, in additional health care costs, or in prolonged clinical examinations. Thus, a quantitative and objectively assessable tool that can provide an overview of the skeletal musculature status, which can be easily included in the routine clinical setting, is urgently needed. Methods that estimate muscle quantity comprise, among others, magnetic resonance imaging (MRI) and computed tomography (CT) scans of the lumbar muscles obtained on abdominal CT scans [8,9,10,11,12,13,14,15,16,17]. However, in the case of brain tumors or other neurological diseases, radiological images of the abdomen are most commonly not routinely available. Recently, TMT, obtained on routinely performed MR images, has been shown to be able to estimate skeletal muscle mass [18]. Therefore, it has been proposed to be a potential parameter by which to identify patients who suffer from sarcopenia. 

The purpose of this study was to further evaluate the anatomical–functional relationship between TMT, obtained on routinely performed cranial MR images, and muscle strength, provided by grip strength measurements, to gain information about muscle quantity and quality of a patient from one single parameter. Furthermore, we aimed to provide the first normal collective database, including sex- and age-related TMT standard values, which will form an important part of the diagnostic pathway. The general aim is to supply healthcare professionals with an independent measure based on routinely assessed, clinically indicated cranial MR images, particularly in patients with neurological disorders who are at risk of sarcopenia, to provide an initial overview of skeletal muscle mass and function. In patients with TMT values below those of the normal collective, further diagnostic examinations will prove or rule out the presence of sarcopenia. 

## 2. Methods 

### 2.1. Study Cohorts

The study is based on two different study cohorts. The retrospective study cohort consisted of two MRI repositories: The Enhanced Nathan Kline Institute-Rockland Sample (NKI-RS); and the Designed Database of MR Brain Images of Healthy Volunteers (MIDAS) [19,20]. Imaging and clinical data from healthy Caucasian individuals were collected and further used to build the retrospective normal collective. 

The results of the retrospective cohort of the healthy volunteers were validated in a prospective, single-center cohort comprising sequentially included Caucasian patients with clinically indicated MRI examinations of the brain, between September 2018 and February 2019, at the Department of Biomedical Imaging and Image-Guided Therapy, Medical University of Vienna, Austria. These patients were subdivided into five sub-cohorts with different disease entities (1. neuro-oncological patients, 2. patients with cerebrovascular disease, 3. patients with demyelinating disorders of the central nervous system, 4. patients with psychiatric disease, and 5. patients with “other disease entities”) for further analysis.

T1-weighted, non-contrast-enhanced MR images were available for all volunteers and patients, respectively. In the prospective cohort, contrast-enhanced, T1-weighted MR images were also available in some patients (if clinically indicated). An overview of the type of functional and clinical data collected is given in Figure 1.

### 2.2. TMT Measurements

TMT was measured on isovoxel (1 × 1 × 1 mm^3^) T1-weighted MR images, perpendicularly to the long axis of the temporal muscle on an axial plane, which was oriented parallel to the anterior commissure-posterior commissure line, as described previously (Figure 2a) [21]. Predefined anatomical landmarks, such as the Sylvian fissure (anterior–posterior orientation) and the orbital roof (craniocaudal orientation) were used to guarantee a high reproducibility of TMT values (represented in Figure 2b,c). An example of TMT measurements in a healthy volunteer on T1-weighted, non-contrast-enhanced brain MR images is provided in Figure 2c. TMT was assessed on both sides in each individual. For further analysis, mean TMT was calculated by summing up those measurements and dividing them by two. All TMT measurements were performed by a board-certified radiologist (main reader = JF). If additional T1-weighted, contrast-enhanced MR images were available, TMT was also assessed by a second radiologist (JL). Temporal muscles indicating any kind of post-therapeutic changes that might have affected their thickness (e.g., muscle edema or atrophy due to craniotomy or radiation therapy) were excluded from further analysis. 

### 2.3. Clinical Examinations 

All clinical examinations were performed immediately before or after the MRI imaging. 

The assessment of hand-grip strength was based on a previously recommended standardized approach [22]. In total, patients underwent three measurements per side. The mean grip strength was calculated for each side and the value of the hand with the highest mean grip strength, referred to as the dominant hand, was used for further statistical analysis. All grip strength measurements were performed using a Lafayette Instrument JAMAR Hydraulic Hand Dynamometer (Model J00105) [22].

The BMI of the patients was assessed by asking the patients their size and current weight. BMI was calculated by dividing a patient´s weight (in kg) by the patient’s height (in meters squared). An unintended weight loss was documented in the event of a reported weight loss > 5 kg within the last year [23]. The existence of temporal wasting was evaluated visually, based on the following criteria: lowering eyelid and eyebrows; skin laxity in the periorbital and cheek area; and changes to the overall face shape [24]. The neurological deficits of the patients were evaluated with the help of the NANO scale, which encompasses nine neurological domains: gait; strength; upper extremity; ataxia; sensation; visual fields; facial strength; language; level of consciousness; and behavior. Each domain was scored based on recently published criteria [25]. A high score indicates a worse neurological function, while a low score represents a good neurological status (score range 0–23). 

### 2.4. Statistical Analysis

Sex- and age-related mean TMT reference values were given as means with standard deviations and ranges. To determine gender-related cutoff values for sarcopenia, mean reference values of healthy young adults (18–40 years) were calculated, and cutoff points were defined as—2.5 standard deviations (SD), as recommended in the literature [2,25]. Inter- and intra-rater agreement were calculated by using an intra-class correlation (ICC; two-way mixed for absolute agreement). The correlation of mean TMT and nonlinear associations was obtained by √R^2^ (age) and by Pearson correlation methods, in the case of linear associations (grip strengths, BMI, and NANO-Scale). To further investigate the impact of age and sex on the correlation of grip strength and TMT, a Pearson correlation was performed, using age and sex as control variables.

An unpaired *t*-test was used to investigate the difference between TMT values in patients with and without clinically suspected temporal muscle wasting and in patients with and without reported unintended weight loss of >5 kg in the last year. Statistical analyses were performed, using the Statistical Package IBM SPSS Statistics for Windows, Version 25.0 (IBM, Armonk, New York, NY, USA). A two-tailed *p*-value of <0.05 was considered statistically significant. 

## 3. Results 

### 3.1. Reference Mean TMT Values

This retrospective normal collective consisted of 624 healthy volunteers (240 males; 384 females), ranging from 18 to 85 years of age. TMT values in the retrospective normal cohort ranged from 3.75 to 15.75 mm. Mean TMT values were significantly higher in male volunteers compared to female healthy volunteers (overall, *p* < 0.001; 18–29 years, *p* < 0.001; 30–39 years, *p* < 0.001; 40–49 years, *p* < 0.001; 50–59 years, *p* < 0.001; 60–69 years, *p* < 0.001; 70–79 years, *p* = 0.011; >80 years, *p* = 0.038). The age- and sex-related mean TMT reference values are listed in Table 1.

### 3.2. Correlation of TMT with Clinical Characteristics in the Retrospective Normal Collective

Overall, 422 healthy volunteers were available for further correlation analysis. Descriptive statistics for the healthy volunteers regarding sex, age, BMI, and grip strength are listed in Appendix A.

Mean TMT values showed a strong correlation with grip strength of the dominant hand in the retrospective normal collective (Pearson correlation coefficient 0.746; *p* < 0.001; Figure 3), as well as in the male subgroup (Pearson correlation coefficient 0.662; *p* < 0.001) and female subgroup, respectively (Pearson correlation coefficient 0.721; *p* < 0.001). 

There was a low negative association of mean TMT and age (R^2^ correlation coefficient 0.20; *p* < 0.001; Figure 4) and no association between mean TMT and BMI values (Pearson correlation coefficient 0.116; *p* = 0.042).

### 3.3. Correlation of TMT with Clinical Patient Characteristics in the Prospective Validation Patient Cohort

To validate the results of the retrospective normal collective, we consecutively included 130 patients with clinically indicated MR examinations of the brain. An overview of the patient characteristics compared to the retrospective normal collective is given in Appendix A.

Even when age was added as a control variable to the correlation between TMT and grip strength, the correlation remained strong in neuro-oncological patients (Pearson correlation coefficient 0.650; *p* < 0.001; Figure 5), patients with demyelinating disease of the central nervous system (Pearson correlation coefficient 0.809, *p* < 0.001), patients with psychiatric disorders (Pearson correlation coefficient 0.669, *p* = 0.024), and in the residual “other disease entities” patient cohort (Pearson correlation coefficient 0.667, *p* < 0.001); and it was moderate in patients with cerebrovascular disease (Pearson correlation coefficient 0.444, *p* = 0.023) listed in Appendix A.

However, when both age and sex were added as control variables, the correlation between TMT and grip strength weakened among all patient subgroups (neuro-oncological patients, Pearson correlation coefficient 0.516, *p* = 0.001; patients with demyelinating disease of the central nervous system, Pearson correlation coefficient 0.557, *p* < 0.048; patients with psychiatric disorders, Pearson correlation coefficient 0.458, *p* = 0.184; the residual “other disease entities” patient cohort, Pearson correlation coefficient 0.567, *p* = 0.001; and with cerebrovascular disease, Pearson correlation coefficient 0.217, *p* = 0.298).

The Pearson correlation revealed no association between mean TMT values and age (Pearson correlation coefficient -0.199; *p* = 0.023) and a low correlation between mean TMT values and BMI (Pearson correlation coefficient 0.227; *p* = 0.009).

In the prospective patient cohort, measures of temporal wasting, unintended weight loss, and neurological deficits obtained by the NANO scale were also performed.

The difference in TMT values between patients with (*n* = 37) and without an unintended weight loss of <5 kg within the last year was not statistically significant (*p* = 0.067); however, after splitting the study population along gender lines, a significant difference in male patients could be revealed (*p* = 0.015), but there was, however, no difference in female patients (*p* = 0.861).

Similarly, in patients with temporal wasting obtained visually (*n* = 33), TMT values were significantly lower (*p* = 0.040) only in male patients. No corresponding findings could be revealed in the female prospective patient cohort (*p* = 0.504). 

The neurological deficits of the patients, assessed by the NANO scale, were also investigated to rule out any interference with the grip strength measurements. In the current study, patients ranged from 0 to 10 points on the NANO scale (mean 1.5, SD 2.1). No correlation was found between hand-grip strength and NANO scale results (Pearson correlation coefficient −0.236).

### 3.4. Assessment of Inter- and Intra-Rater Agreement 

Intra-rater reliability between TMT measurements on contrast-enhanced and non-contrast-enhanced T1-weighted MR images was assessed in the prospective validation cohort (*n* = 89), resulting in a next-to-perfect intra-rater agreement (ICC = 0.945). Moreover, inter-rater agreement was calculated for the TMT measurements on T1-weighted contrast-enhanced MR images, in the prospective validation cohort (*n* = 89), which also showed a very high agreement (ICC = 0.912).

## 4. Discussion

### 4.1. Correlation of TMT and Grip Strength

We investigated an anatomical–functional association between the thickness of the temporal muscle and grip strength, which is currently reported to be the most reliable assessment of muscle function, to investigate the potential of TMT to provide information about the physical condition of an individual [2]. TMT and the grip strength of the dominant hand showed a strong correlation in healthy volunteers (ρ = 0.746; *p* < 0.001), as well as in patients with various neurological disorders (ρ = 0.649; *p* < 0.001). These data provide evidence that TMT is a useful parameter with which to also estimate the strength of the skeletal musculature besides skeletal muscle quantity [18]. We could further demonstrate that there is nearly no impact of age on the correlation between TMT and grip strength in the patient cohort; however, the correlation of TMT and grip strength tended to weaken among all sub-cohorts of patients after correcting for sex and age. It can thus be deduced that the impact of age on the correlation of TMT and grip strength is variable, whereas the correlation of TMT and grip strength between male and female patients is slightly different. This gender-related difference is of high interest and importance and should be further investigated with regard to patients’ outcome in clinical trials focused on different specific disease entities.

### 4.2. Reference Values and Gender-Related TMT Cutoff Points

Furthermore, we provided sex- and age-related reference values for TMT measurements obtained on cranial MR images, in a retrospective normal collective ranging from 18 to >80 years, including 624 healthy volunteers (male, *n* = 240; female, *n* = 384). Based on these data, sex-related cutoff points for the diagnosis of sarcopenia were calculated as defined by the updated EWGSOP recommendations, using normative references (healthy young adults) between 18 and 40 years [2]. Sex-related cutoff points were set at 2.5 standard deviations (SD) below the normative references, respectively, resulting in cutoff points to diagnose sarcopenia of 6.3 mm for male patients and 5.2 mm for female patients, respectively. 

### 4.3. Correlation of TMT with Further Clinical Parameters 

As previously shown, TMT was significantly lower in female individuals compared to males, within each study population (overall; *p* < 0.001) [17]. Although muscle mass and strength are known to decline with age, there was only a low negative association between mean TMT and age in the healthy volunteers (R^2^ correlation coefficient 0.20; *p* < 0.001), and no significant correlation in patients who suffered from various neurological symptoms (Pearson correlation coefficient −0.199; *p* = 0.023). These results reflect the hypothesis of a difference between chronological age and biological age, including, among others, information about an individual’s physical condition, which may be more meaningful, with regard to clinical decision-making, than considering the chronological age of a patient alone [26,27]. Furthermore, there was no correlation between BMI and TMT in the normal collective individuals and only a low correlation (Pearson correlation coefficient 0.227; *p* = 0.009) in the prospective neurological patients. One explanation could be that the information about weight and height were self-reported, and thus, prone to reporting bias, due to the tendency of people to overestimate their height and underestimate their weight. However, it could also be due to the inability of the BMI (comprising the weight and height of an individual) to differentiate between fat and lean body mass compared to TMT, which has been shown to be a reliable parameter by which to estimate skeletal muscle mass [17]. 

A significant difference between patients with and without visually determined temporal wasting was found only in male patients (*p* = 0.040). The appearance of the temples comprises various structures: the temporal muscle; the temporal fat pad; and the skull shape. Thus, significantly thicker temporal muscles in male patients may be the reason for the higher impact of the reduction of TMT on the visually determined temporal wasting. Moreover, the visual evaluation of temporal wasting is a very subjective judgment that is highly dependent on the performing physician. 

Similarly, there was a significant difference in TMT values only in male patients, with and without unintended weight loss > 5 kg in the last year (*p* = 0.015). This was also the case for grip strength, where the only significant differences between patients with and without unintended weight loss were also revealed in only male patients (*p* = 0.012). One limiting factor of this result is that the weight loss of >5 kg in the last year was not measured clinically, but relied on self-reported information. In addition, Cosway et al. found no correlation of hand-grip strength and treatment-related weight loss in head/neck cancer patients and suggested that hand-grip strength reflects functional, rather than nutritional, status [28]. In their study, the authors did not differentiate between male and female patients; thus, perhaps positive effects in the male patient cohort were missed. Based on our results, we hypothesize that unintended weight loss in male patients may be a marker for male muscle-mass loss rather than muscle mass loss in female patients. However, this assumption has to be investigated in further clinical studies. 

The impact of neurological deficits on the grip-strength measurements in the current study was ruled out, using the NANO scale.

### 4.4. TMT Measurements

Compared to the currently used techniques to obtain muscle quantity, the TMT measurement procedure is simple, fast, reliable, independent of the patient’s phenotype, and no additional examinations are needed. TMT can be obtained on T1-weighted MR images, with and without contrast enhancement (intra-rater agreement, ICC = 0.945), on routinely performed cranial MR images. Using predefined anatomical landmarks, the inter-rater reliability was excellent in the current study (ICC = 0.912), as well as in previous ones [18,21]. The measurement of TMT in one patient takes approximately 30 seconds [18]. This may help to integrate the determination of muscle quantity into clinical practice in neurological patients who undergo a clinically indicated MRI of the brain. 

The use of TMT, among other craniofacial muscles, as a surrogate parameter for the diagnosis of sarcopenia is justified for several reasons. The temporal muscle is one of very few muscles that can usually be delineated in its whole extent on MR images of the brain. This is especially important in patients with suspected post-therapeutic muscle changes, such as muscle edema or atrophy, due to previous interventions. Furthermore, TMT has been previously shown to have a prognostic role in different diseases. Lisiecki et al. revealed a correlation between TMT with ventilator and hospital days in trauma patients [29]. Katsuki et al. were able to predict the outcome after aneurysm treatment in patients over 75 years of age with an aneurysmal subarachnoid hemorrhage [30]. Moreover, TMT showed a high correlation with overall survival in patients with brain metastasis from melanoma, breast, and lung cancer, as well as in patients with progressive glioblastoma [21,31,32]. Furthermore, temporal muscle volume was used to predict the in-hospital stay of children with non-syndromic craniosynostosis, which is, again, more time-consuming, and therefore unsuitable for integration into the routine clinical setting [33].

### 4.5. Adapted Diagnostic Algorithm with Which to Identify Neurological Patients at Risk for Sarcopenia

On the basis of the current findings, we suggest that TMT should be assessed routinely on clinically indicated cranial MR images, especially in patients with neurological disorders at risk for sarcopenia. The thickness of the temporal muscle should be used to provide an initial overview of the skeletal muscle mass and strength, without replacing other diagnostic procedures. Once patients present with TMT values under the sex-related cutoff points, further tests would be necessary to confirm or rule out the presence of sarcopenia, as stated in the EWGSOP Find-Assess-Confirm-Severity (FACS) recommendation, using the “Assessment” and “Confirmation” procedures (see Figure 6). Another advantage is that, in some neurological disorders, cranial MRIs are obtained on a regular basis (e.g., brain tumor patients or patients with demyelinating disease of the central nervous system, specifically, patients with multiple sclerosis) to monitor the disease course. In these cases, TMT can be used to assess the musculature status of the patients provided by longitudinal MRI examinations and uncover patients whose condition worsens over the course of their disease or who newly develop sarcopenia. This is why we suggest implementing TMT in the updated EWGSOP FACS recommendations at the level of “Find” cases, in addition to the already implemented criteria, such as “clinical symptoms” and “SARC-F questionnaire” if a cranial MRI examination is already available due to clinically indicated reasons (see Figure 6).

### 4.6. Limitations

Although the results of this study are based on a large retrospective database and were validated in an independent prospective patient cohort, our study faces some limitations. The determined sex-related cutoff points of -2.5 SD below the normative references are based on the literature. Their impact on predefined clinical endpoints, as well as the impact of the severity of the disease on the correlation of TMT and grip strength, must be further validated in clinical trials with different disease entities. Moreover, the results of this study are based exclusively on a Caucasian population. Therefore, further studies are needed in order to establish reference values and cutoff points for people of different ethnic backgrounds, to reflect possible variations between ethnicities. The temporal muscle has the advantage that it can be depicted in its entirety on cranial MR images, which is of great importance to rule out any alterations that could influence the muscle thickness. However, it should be mentioned that the diameter of this muscle is relatively small, and it is, therefore, of the utmost importance to adhere strictly to the predefined anatomic landmarks and use high-resolution MRI images to reduce partial volume artefacts and provide high measurement accuracy. Furthermore, TMT could be influenced by oral or dental disease [34]. To overcome this problem, TMT values were measured on both sides and divided by two, to calculate the mean TMT value for each patient and reduce dental- or oral-related muscle changes as much as possible. 

Future studies should investigate the correlation of TMT with relative and absolute total body muscle mass, and further clinical trials will need to define the relation of nutritional parameters on TMT and muscle strength as a basis for specific interventions aimed at improving the physical function of high-risk patient populations.

## 5. Conclusions

Within the scope of this study, we introduced TMT as a parameter which is reliable, fast, and objectively assessable on routine cranial MR images and shows a high correlation to muscle strength in healthy individuals, as well as in patients with different neurological disorders. As TMT has also been previously shown to be a useful parameter for the estimation of skeletal muscle mass, we suggest integrating TMT into the diagnostic workup of sarcopenia assessment in neurological patients with clinically indicated cranial MR images. 

## Figures and Tables

**Figure 1 jcm-09-01272-f001:**
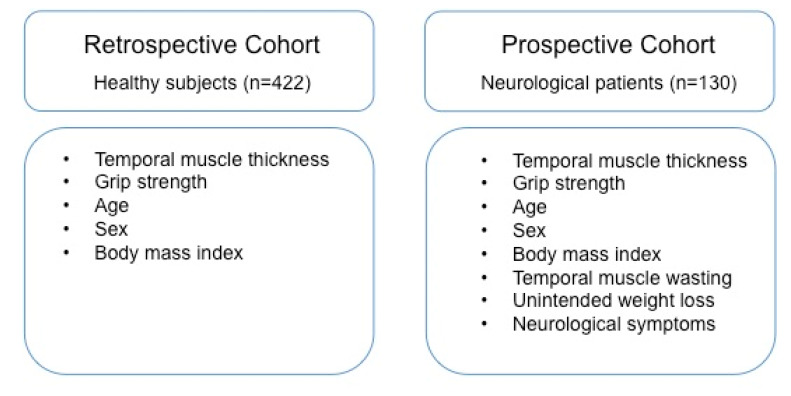
Overview of the type of clinical and functional data collected. The study was approved by the ethics committee of the Medical University of Vienna (1406/2017).

**Figure 2 jcm-09-01272-f002:**
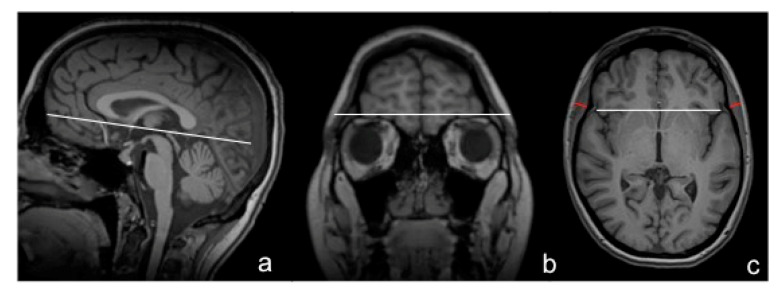
Anatomical landmarks represented with white lines (**a**–**c**) and an example of a TMT measurement in a healthy volunteer on T1-weighted, non-contrast-enhanced cranial MR images depicted in red (**c**).

**Figure 3 jcm-09-01272-f003:**
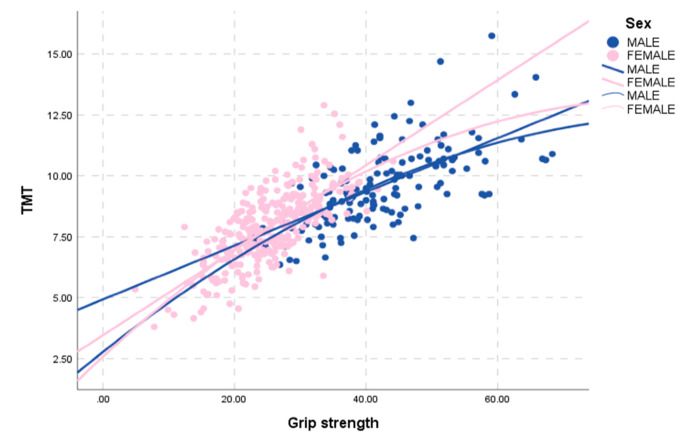
Correlation between mean TMT values and grip strength of male (blue) and female (pink) healthy volunteers.

**Figure 4 jcm-09-01272-f004:**
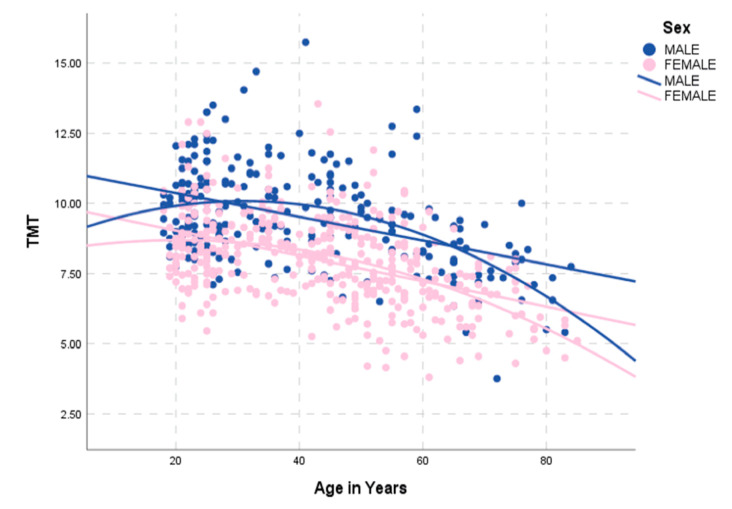
Correlation between age and mean TMT values in male (blue) and female (pink) healthy volunteers.

**Figure 5 jcm-09-01272-f005:**
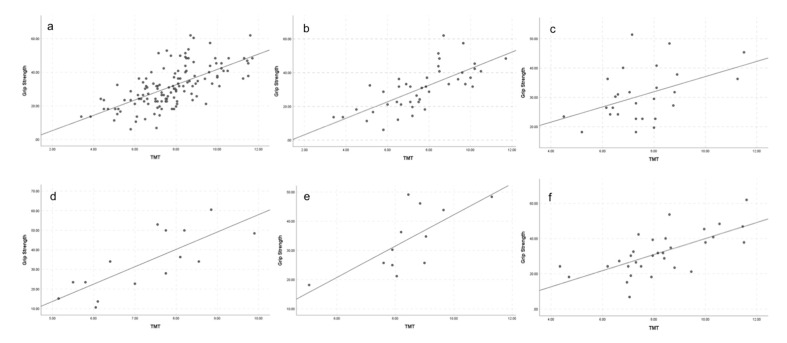
Correlation between mean TMT values and grip strength in the overall patient population (**a**) and subdivided into the different disease entities of neuro-oncological patients (**b**), patients with cerebrovascular disease (**c**), patients with demyelinating disease of the central nervous system (**d**), patients with psychiatric disorders (**e**), and patients with “other disease entities” (**f**).

**Figure 6 jcm-09-01272-f006:**
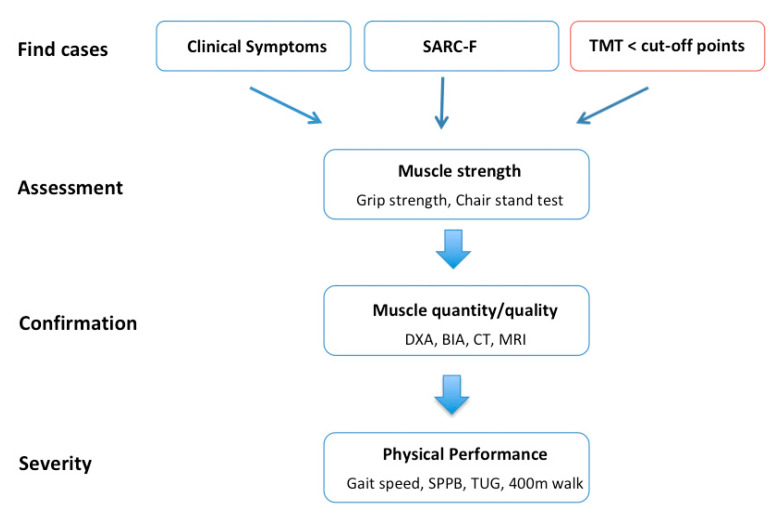
Adapted algorithm for sarcopenia case identification, diagnosis, and quantification of severity in patients with neurological disorders (modified after Cruz-Jentoft et al., 2019) [2]. Abbrevations: BIA: bioelectrical impedance analysis; CT: computer tomography; DXA: dual energy x-ray; MRI: magnet resonance imaging; SARC-F: sarcopenia questionnaire; SPPB: short physical performance battery; TMT: temporal muscle thickness; TUG: time up and go.

**Table 1 jcm-09-01272-t001:** Age- and sex-related mean TMT reference values, minimum and maximum TMT values, and SD per subgroup.

SEX	Age-Group		*n*	Minimum (mm)	Maximum (mm)	Mean (mm)	SD
**MALE**	18–29	TMTmean	98	7.10	13.50	9.9709	1.42971
	30–39	TMTmean	39	7.35	14.70	9.9654	1.60813
	40–49	TMTmean	35	6.65	15.75	10.0300	1.71679
	50–59	TMTmean	26	6.50	13.35	9.4769	1.63139
	60–69	TMTmean	24	5.40	9.80	7.9958	1.06157
	70–79	TMTmean	13	3.75	10.00	7.6846	1.49086
	80+	TMTmean	5	5.40	7.75	6.5100	1.06031
**FEMALE**	18–29	TMTmean	108	5.45	12.90	8.6167	1.44770
	30–39	TMTmean	54	6.45	11.25	8.6806	1.16309
	40–49	TMTmean	77	5.25	13.55	8.2065	1.43282
	50–59	TMTmean	77	4.15	11.90	7.5916	1.53549
	60–69	TMTmean	43	3.80	9.70	6.6570	1.22200
	70–79	TMTmean	20	4.30	8.10	6.5075	1.03151
	80+	TMTmean	5	4.50	5.85	5.1700	1.57511

Abbreviations: mm: millimeter; SD: standard deviation; TMT: temporal muscle thickness.

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
