# Peer review of "Sarcopenia in Neurological Patients: Standard Values for Temporal Muscle Thickness and Muscle Strength Evaluation"

_jcm, 2020, doi:10.3390/jcm9051272_

Round 1

Reviewer 1 Report

Dear editor,

The following review comments pertain to paper "Sarcopenia in neurological patients: standard values 3 for temporal muscle thickness and muscle strength 4 evaluation”. I have reviewed the submission.

The authors suggest that Temporal muscle thickness should be assessed routinely on clinically indicated cranial MR images, especially in patients with neurological disorders at risk of sarcopenia.

The paper is well written and interesting. The authors provide a useful report that may aid the further understanding of assessment and treatment of  sarcopenia. I have a few points, for the authors to consider.

SPECIFIC COMMENTS.

Page 4  Line  116 « The BMI of the patients was assessed by asking the patients their size and current weight»

Height and weight should be measured, if possible. Could you please discuss the results. As height and weight are self-reported, they may be prone to reporting bias. Individuals, on average, tend to overestimate their height and underestimate their weight.

Page 4  Line  121.    Please  Add reference.

Page 4. Line 129,130. Inter- and intra-rater agreement  were calculated using an intra-class correlation (ICC:)”.

The authors should additionally state which correlation coefficient model was used.

Page 5. Line 153. Please add dot.

Author Response

Reviewer 1

The following review comments pertain to paper "Sarcopenia in neurological patients: standard values  for temporal muscle thickness and muscle strength evaluation”. I have reviewed the submission. The authors suggest that Temporal muscle thickness should be assessed routinely on clinically indicated cranial MR images, especially in patients with neurological disorders at risk of sarcopenia.

The paper is well written and interesting. The authors provide a useful report that may aid the further understanding of assessment and treatment of sarcopenia. I have a few points, for the authors to consider.

SPECIFIC COMMENTS.

Page 4  Line  116 « The BMI of the patients was assessed by asking the patients their size and current weight»

Height and weight should be measured, if possible. Could you please discuss the results. As height and weight are self-reported, they may be prone to reporting bias. Individuals, on average, tend to overestimate their height and underestimate their weight.

Thank you for the important comment. To highlight this fact, we added the following sentence in the “Discussion” section:

Furthermore, there was no correlation between BMI and TMT in the normal collective individuals and only a low correlation (Pearson correlation coefficient 0.227; p = 0.009) in the prospective neurological patients. One explanation could be that the information about weight and height were self-reported and thus, prone to reporting bias, due to the tendency of people to overestimate their height and underestimate their weight.However, it could also be due to the inability of the BMI (comprising the weight and height of an individual) to differentiate between fat and lean body mass compared to TMT, which has been shown to be a reliable parameter by which to estimate skeletal muscle mass.20

Page 4  Line  121.    Please  Add reference.

We added the following reference:

Coleman, S. R. & Grover, R. The anatomy of the aging face: Volume loss and changes in 3-dimensional topography. Aesthetic Surg. J.(2006). doi:10.1016/j.asj.2005.09.012

Page 4. Line 129,130“Inter- and intra-rater agreement  were calculated using an intra-class correlation (ICC:)”.

The authors should additionally state which correlation coefficient model was used.

To specify we added the following description:

Inter- and intra-rater agreement were calculated using an intra-class correlation (ICC; two-way mixed for absolute agreement).

Page 5. Line 153. Please add dot.

We are sorry for the mistake and completed the sentence as follows:

Descriptive statistics for the healthy volunteers regarding sex, age, BMI, and grip strength are listed in Supplementary Table 1.

Reviewer 2 Report

A good study reporting the results of the evaluation between temporal muscle thickness and muscle strength evaluation. However, authors should consider number of covariate factors to analyze the correlation between temporal muscle thickness and muscle strength as below.

First, adjusted parameters such as tooth condition, periodontal disease, and denture that affect temporal muscle must be considered.

Second, the patient's nutritional parameters should also be adjusted for assessing the effect of temporal muscle on muscle strength.

Third, you need to measure the variables that can assess your total body muscle mass. And an analysis that considers the relative muscle mass compared to the total body mass is needed.

This study seems to require significant revisions to the above statistical aspects and the design of the study. Therefore, I think it will be difficult to publish the paper to JCM with the design of the research.

Author Response

A good study reporting the results of the evaluation between temporal muscle thickness and muscle strength evaluation. However, authors should consider number of covariate factors to analyze the correlation between temporal muscle thickness and muscle strength as below.

First, adjusted parameters such as tooth condition, periodontal disease, and denture that affect temporal muscle must be considered.

Thank you for this comment. We are aware of this problem, therefore, we assess the temporal muscle thickness on both sides and use the mean TMT for the further calculation to reduce dental- or oral-related muscle changes as far as possible. To clarify, we added the following paragraph in the Discussion section:

Furthermore, TMT could be influenced by oral or dental disease.41To overcome this problem, TMT values were measured on both sides and divided by two to calculate the mean TMT value for each patient and reduce dental- or oral- related muscle changes as much as possible.

Further, we added the following reference:

  1. Grunheid T., et al. The adaptive response of jaw muscles to varying functional demands. Eur J Orthod. (2009). doi:10.1093/ejo/cjp093

Second, the patient's nutritional parameters should also be adjusted for assessing the effect of temporal muscle on muscle strength.

We agree that investigation of nutritional parameters and their influence on TMT and muscle strength are of interest and added the following sentence:

“Further clinical trials will need to define the relation of nutritional parameters on TMT and muscle strength as a basis for specific interventions that are aimed at improving the physical function of high-risk patient populations.”

Third, you need to measure the variables that can assess your total body muscle mass. And an analysis that considers the relative muscle mass compared to the total body mass is needed.

We agree that a correlation of TMT with different body compartments, especially relative muscle mass, is important in defining temporal muscle thickness as a diagnostic tool for sarcopenia.

However, we do not consider temporal muscle thickness alonewhen diagnosing sarcopenia. In our manuscript, we suggest using temporal muscle thickness as an additionalparameter to detect patients who may be at risk for sarcopenia.

The predefined goal of our study was to assess the correlation of TMT with grip strength, as an established surrogate parameter for physical functioning, and the study was designed accordingly.Based on the recommendations of the revised European consensus on the definition and diagnosis of sarcopenia (Cruz-Jentoftet al. 2019), the twokey features, when it comes to the diagnoses of sarcopenia, are muscle quantity(obtained by BIA, DXA, or CT/MRI of the lumbar skeletal muscle mass) and muscle quality(obtained by grip strength, chair stand test).

Temporal muscle thickness has been previously shown to provide a high correlation with the lumbar skeletal muscle mass obtained on abdominal CT scans (Ranganathanet al., 2014; Leitner et al., 2018), representing muscle quantity. In the current study, we could show a high correlation between temporal muscle thickness and grip strength measurements, representing muscle quality. Based on these results, we suggest using temporal muscle thickness as a tool with which to provide an indication of the loss of muscle mass and/or function. Moreover, temporal muscle thickness measurements should not replace the established “Case finding” tools, such as “Clinical Symptoms” or the “SARC-F questionnaire.” It should be seen as a screening procedure, which can raise the awareness of potential sarcopenia signs in patients with clinically indicated MRI examinations of the brain, which results in no additional costs, radiation exposure, or time.

If a patient presents with a temporal muscle thickness below 2.5 standard deviations of the sex-related reference values (also given in this manuscript), it is recommended that further diagnostic procedures be initiated, such as those recommended in the revised European consensus on the definition and diagnosis of sarcopenia (Cruz-Jentoftet al. 2019), including: the SARC-F questionnaire; a grip strength examination; and DEXA/BIA measurements, etc.

To correct potential misleading statements, we rephrased the following sentences in the Abstract:

“Temporal muscle thickness (TMT) was investigated as a novel surrogate marker on MRI examinations of the brainto detect patients who may be at risk for sarcopenia.”

Further, we added this sentence in the “Discussion” section:

“The thickness of the temporal muscle should be used to provide an initial overview of the skeletal muscle mass and strength without replacing other diagnostic procedures.”

We agree that further research questions arise from our data and the following sentence was added to the manuscript:

“Future studies should investigate the correlation of TMT with relative and absolute total body muscle mass.”

This study seems to require significant revisions to the above statistical aspects and the design of the study. Therefore, I think it will be difficult to publish the paper to JCM with the design of the research.

We thank the reviewer for pointing out relevant questions for follow-up studies, but do believe that the design of our study is suitable to provide relevant new information and make meaningful conclusions. We have highlighted some areas of further research in our revised manuscript.

Reviewer 3 Report

In this manuscript, the authors aimed at assessing the prospective evaluation of TMT as a diagnostic factor to evaluate sarcopenia in neurological disorders. TMT analysis could represent an innovative and reliable tool to evaluate skeletal muscle mass in these patients. The findings of this work are interesting and have the merit to propose an easy and low-cost method that could be integrated in the diagnostic protocol to examine sarcopenia in patients suffering from neurological disorders. However, there are some concerns that should be addressed:

  • Figure 3 is not properly cited in the main text.
  • The cohort of patients (including the subgroups of disease entities) should be better described in the "material and methods" section.
  • It seems that correlation analysis are not stratified for severity/stage of each pathological condition. This limitation should be addressed or at least discussed in the manuscript.
  • Correlations between TMT and overall patient cohort (as well as between TMT and each disease entity) should be shown as scatter plot graphs, similarly to fig. 3 and 4. 
  • It would be interesting to clearly see the prospective effect of age and sex on TMT/grip strength correlation for each disease entity. In my opinion, it is not easy to obtain this information along the manuscript. 

Author Response

Reviewer 3

In this manuscript, the authors aimed at assessing the prospective evaluation of TMT as a diagnostic factor to evaluate sarcopenia in neurological disorders. TMT analysis could represent an innovative and reliable tool to evaluate skeletal muscle mass in these patients. The findings of this work are interesting and have the merit to propose an easy and low-cost method that could be integrated in the diagnostic protocol to examine sarcopenia in patients suffering from neurological disorders. However, there are some concerns that should be addressed:

  • Figure 3 is not properly cited in the main text.

We apologize for this mistake and have added the missing citation in the text as follows:

            Mean TMT values showed a strong correlation with grip strength of the dominant hand in the           retrospective normal collective (Pearson correlation coefficient 0.746; p<0.001; Figure 3),

  • The cohort of patients (including the subgroups of disease entities) should be better described in the "material and methods" section.

The following paragraph was added to the “Materials and Methods” section:

The results of the retrospective cohort of the healthy volunteers were validated in a prospective single-center cohort comprising sequentially included Caucasian patients with clinically indicated MRI examinations of the brain between September 2018 and February 2019 at the Department of Biomedical Imaging and Image-guided Therapy, Medical University of Vienna, Austria. These patients were subdivided into five sub-cohorts with different disease entities (1. neuro-oncological patients, 2. patients with cerebrovascular disease, 3. patients with demyelinating disorders of the central nervous system, 4. patients with psychiatric disease, and 5. patients with “other disease entities”) for further analysis.

  • It seems that correlation analysis are not stratified for severity/stage of each pathological condition. This limitation should be addressed or at least discussed in the manuscript.

This is correct. There was no stratification of patients for severity/stage of their disease. The prospective cohort was included in this study to show that the correlation between grip strength and temporal muscle thickness of healthy volunteers is also demonstrable in sick patients independent of the disease entity or the severity/stage of disease. However, this is, of course, an important clinical question and should be addressed in further clinical trials.

To highlight this point, we added the following sentence in the “Discussion – Limitations” section:

Although the results of this study are based on a large retrospective database and were validated in an independent prospective patient cohort, our study faces some limitations. The determined sex-related cut-off points of -2.5 SD below the normative referencesare based on the literature. Their impact on predefined clinical endpoints, as well as the impact of the severity of the disease on the correlation of TMT and grip strength must be further validated in clinical trials with different disease entities.

  • Correlations between TMT and overall patient cohort (as well as between TMT and each disease entity) should be shown as scatter plot graphs, similarly to fig. 3 and 4. 

To visualize the correlation of TMT and grip strength in the prospective study we added Figure 5 in the Results section, which depicts the correlation of the whole prospective study group (a) and of each disease entity (b-f).

Figure 5: Correlation between mean TMT values and grip strength of the overall patient population (a) and subdivided into the different disease entities of neuro-oncological patients (b), patients with cerebrovascular disease (c), patients with demyelinating disease of the central nervous system (d), patients with psychiatric disorders (e), and “other disease entities” (f).

  • It would be interesting to clearly see the prospective effect of age and sex on TMT/grip strength correlation for each disease entity. In my opinion, it is not easy to obtain this information along the manuscript.

To reveal the impact of age and sex on the TMT/grip strength correlation for each disease entity we used a Pearson correlation for TMT and grip strength and added age, as well as sex as control variables. Therefore, the following sentence has been added to the “Methods”:

To further investigate the impact of age and sex on the correlation of grip strength and TMT, a Pearson correlation was performed using age and sex as control variables.

The results are presented in the “Results” section as follows:

Even when age was added as a control variable to the correlation between TMT and grip strength, the correlation remained strong in neuro-oncological patients (Pearson correlation coefficient 0.650; p<0.001), patients with demyelinating disease of the central nervous system (Pearson correlation coefficient 0.809, p<0.001), patients with psychiatric disorders (Pearson correlation coefficient 0.669, p=0.024), and in the residual “other disease entities” patient cohort (Pearson correlation coefficient 0.667, p<0.001), and was moderate in patients with cerebrovascular disease (Pearson correlation coefficient 0.444, p=0.023).

However, when both age and sex were added as control variables, the correlation between TMT and grip strength weakened among all patient subgroups (neuro-oncological patients, Pearson correlation coefficient 0.516, p=0.001; patients with demyelinating disease of the central nervous system, Pearson correlation coefficient 0.557, p<0.048; patients with psychiatric disorders, Pearson correlation coefficient 0.458, p=0.184; the residual “other disease entities” patient cohort, Pearson correlation coefficient 0.567, p=0.001; and with cerebrovascular disease, Pearson correlation coefficient 0.217, p=0.298).

The results are discussed in the “Discussion” section as follows:

Correlation of TMT and grip strength

We investigated an anatomical-functional association between the thickness of the temporal muscle and grip strength, currently the most reliable assessment of muscle function, to investigate the potential of TMT to provide information about the physical condition of an individual 2. TMT and the grip strength of the dominant hand showed a strong correlation in healthy volunteers (ρ= 0.746; p<0.001), as well as in patients with various neurological disorders(ρ= 0.649; p<0.001). These data provide evidence that TMT is a useful parameter with which to also estimate the strength of the skeletal musculature in addition to skeletal muscle quantity.18

We could further demonstrate that there was nearly no impact of age on the correlation of TMT and grip strength in the patient cohort; however, the correlation of TMT and grip strength tended to weaken among all sub-cohorts of patients after correcting for sex and age. It can thus be deduced that the impact of age on the correlation of TMT and grip strength is variable, whereas the correlation of TMT and grip strength between male and female patients is slightly different. This gender-related difference is of high interest and importance and should be further investigated with regard to patients’ outcome in clinical trials focused on different specific disease entities.

Round 2

Reviewer 2 Report

I accepted this article due to many of revisions. 

Reviewer 3 Report

The manuscript looks significantly improved if compared to the original version, as the authors properly addressed all the reviewer's concerns, and provided an exhaustive and satisfactory point-by-point reply.